# Intravitreal injections of corticosteroid and the risk of central serous chorioretinopathy

**Jae Ryong Song[1,2], Se Joon Woo◯[2]\***

**1** Department of Ophthalmology, Seoul National University College of Medicine, Seoul National University Hospital, Seoul, Republic of Korea, **2** Department of Ophthalmology, Seoul National University College of Medicine, Seoul National University Bundang Hospital, Seongnam, Republic of Korea

\* sejoon1@snu.ac.kr

## Abstract

Central serous chorioretinopathy (CSC) is strongly associated with systemic corticosteroid use, but the risk following intravitreal steroid injections remains unclear, with limited evidence from isolated case reports. This study evaluated the association between intravitreal steroid injections and CSC development using a dual-approach design with Clinical Data Warehouse (CDW) analysis and choroidal thickness validation. We performed CDW analysis of 2,336 patients receiving intravitreal triamcinolone acetonide or dexamethasone implant over 21 years (April 2003–October 2024), screening for CSC within 6 months post-injection. Additionally, we conducted detailed subfoveal choroidal thickness analysis in 269 eyes (January 2023–October 2024) at baseline, 1, 3, and 6 months post-injection using optical coherence tomography. No confirmed CSC cases occurred in either cohort (0/2,336 patients, 95% CI: 0–0.13%; 0/269 eyes). Contrary to the choroidal thickening characteristic of CSC, both steroid formulations produced significant choroidal thinning at 1 month: dexamethasone implant (227.0±65.4μm to 217.9±62.0μm, p<0.0001) and triamcinolone acetonide (219.8±78.7μm to 218.6±77.7μm, p<0.0001). Only 4.9% of eyes showed ≥10% choroidal thickening, and none developed CSC features. Both formulations achieved significant macular edema reduction through 6 months. Intravitreal corticosteroids showed no confirmed CSC cases over two decades with choroidal thinning rather than the thickening characteristic of CSC, suggesting distinct pathophysiological mechanisms and a favorable safety profile regarding CSC development.

## Introduction

Central serous chorioretinopathy (CSC) is a retinal disease characterized by the accumulation of serous fluid beneath the neurosensory retina, particularly in the macular region [1]. This condition manifests with symptoms such as blurred or distorted vision, metamorphopsia, and reduced visual acuity, significantly impacting patients' quality of life [2,3]. CSC predominantly affects middle-aged men, with a multifactorial

**Data availability statement:** The datasets generated and/or analyzed during the current study contain sensitive patient information and cannot be shared publicly due to patient privacy and confidentiality restrictions under Korean personal information protection laws. Data are available from the corresponding author (contact: sejoon1@snu.ac.kr) and the Institutional Review Board of Seoul National University Bundang Hospital (email: snubhirb@gmail.com; Tel: +82-31-787-8801, IRB No. B-2410-932-104) for researchers who meet the criteria for access to confidential data, upon reasonable request and following approval of a data sharing agreement. Data will be maintained securely at Seoul National University Bundang Hospital for the duration required by institutional and regulatory policies.

**Funding:** This work was supported by the Technology Innovation Program (RS-2024-00507933, received by Se Joon Woo) funded by the Ministry of Trade, Industry & Energy (MOTIE, Korea) and by a grant of Korean ARPA-H Project through the Korea Health Industry Development Institute (KHIDI), funded by the Ministry of Health & Welfare, Republic of Korea (RS-2025-25454860, received by Se Joon Woo). The funding organization had no role in the design or conduct of this study.

**Competing interests:** The authors have declared that no competing interests exist.

etiology including psychological stress, genetic predisposition, corticosteroid use, and underlying systemic conditions [3–5]. Moreover, recent studies have shown that the association between CSC and psychosocial factors is phase-dependent, with active CSC patients demonstrating significantly higher stress and depression scores compared to controls [6].

Among various steroid formulations, systemic corticosteroids are acknowledged to be an important risk factor for CSC. In a prospective case-control study by Carvalho-Recchia et al., 52% of CSC patients had a history of exogenous steroid use compared to 18% in the control group, representing a statistically significant difference [4]. Moreover, other routes have also been reported in association with CSC, such as intranasal, intra-articular, topical dermal, inhaled, and in some cases ocular use [7–9].

Intravitreal steroid injections are widely utilized therapeutic interventions for managing various retinal and choroidal diseases, including diabetic macular edema, retinal vein occlusion, and uveitic macular edema [10]. However, their potential role in inducing CSC has not been comprehensively evaluated, with available evidence primarily limited to isolated case reports, including CSC following intravitreal triamcinolone acetonide [11,12] and intravitreal dexamethasone implants [13,14]. While these case reports have raised clinical concerns, this lack of comprehensive research poses significant challenges in determining the true association between intravitreal steroid injection and CSC development.

To address this knowledge gap, this study aims to systematically evaluate the relationship between intravitreal steroid injections and CSC development using a comprehensive dual approach combining large-scale clinical data analysis and detailed imaging evaluation. This research seeks to provide real-world evidence for enhancing clinical decision-making protocols and establishing evidence-based guidelines for the appropriate use of intravitreal corticosteroids.

## Methods

This retrospective observational study was conducted at Seoul National University Bundang Hospital. The study was approved by the Institutional Review Board (No. B-2410-932-104) and conducted in accordance with the principles of the Declaration of Helsinki. Due to the retrospective nature of the study, the requirement for informed consent was waived. Medical records were accessed between January 1, 2025 and April 30, 2025 for research purposes. During data collection, investigators had access to identifiable patient information to ensure accurate clinical data extraction, which was subsequently de-identified and anonymized for analysis to protect patient privacy.

The research utilized two complementary retrospective approaches designed to overcome the limitations inherent in each method when used alone. The Clinical Data Warehouse (CDW) analysis provided comprehensive surveillance across all patients over an extended 21-year period to identify all potential CSC cases but relied on clinical documentation and could not perform detailed optical coherence tomography (OCT) analysis on historical data. The detailed retrospective analysis offered objective validation through choroidal thickness measurements in recent cases with

high-quality OCT data, based on the rationale that CSC is characteristically associated with choroidal thickening; therefore, if intravitreal steroids induce CSC, choroidal thickening should be observed in treated eyes. This dual approach allowed us to combine broad population-level screening with rigorous imaging-based validation.

## Clinical Data Warehouse (CDW) analysis

The Clinical Data Warehouse (CDW) is an institutional electronic database that systematically integrates and stores comprehensive patient information from electronic medical records [15]. The CDW of Seoul National University Bundang Hospital was utilized to identify patients who received intravitreal steroid injections(triamcinolone acetonide and dexamethasone implant) between April 1, 2003, and October 31, 2024. The CDW system contains both structured data and unstructured clinical text data extracted from electronic medical records. Patients who received two types of intravitreal steroid formulations (triamcinolone acetonide and dexamethasone implant) in the ophthalmology department were identified through procedure codes and medication records. These patients were then screened for CSC diagnosis following steroid injection using both structured diagnosis codes containing 'Central serous chorioretinopathy' and unstructured free-text clinical notes containing CSC-related terms ('CSC', 'CSCR', 'central serous chorioretinopathy' and Korean equivalents) within 6 months post-injection based on temporal patterns reported in previous case reports [11–14] and pharmacokinetic studies of dexamethasone implants showing sustained drug release for up to 6 months with negligible residual intraocular steroid thereafter [16,17]. Given that all intravitreal steroid injections at our institution are performed by retinal specialists in dedicated vitreoretinal clinics, and these patients undergo routine post-injection OCT examinations, any development of subretinal fluid or CSC-like features would be identified during these examinations and documented in either diagnosis codes or clinical notes. Only the three suspected cases meeting both injection and diagnosis codes or free-text clinical notes underwent comprehensive manual review by qualified ophthalmologists (JRS, SJW) to determine the temporal and causal relationship between steroid injection and CSC development. CSC was defined as subretinal fluid on optical coherence tomography in the macular region without alternative explanation. When available, fluorescein angiography findings such as focal leakage points provided additional diagnostic confirmation. Moreover, temporal relationships with steroid injections were evaluated. Discrepancies were resolved through consensus discussion (Fig 1).

## Detailed retrospective analysis – Choroidal thickness validation

A detailed retrospective analysis was conducted on patients who received intravitreal steroid injections with either triamcinolone acetonide 40 mg (Maqaid, Hanmi Pharm., Seoul, Korea) or dexamethasone implant 700 mcg (Ozurdex, Korea AbbVie Ltd., Seoul, Korea) between January 2023 and October 2024. This analysis was designed to validate CDW findings through objective choroidal thickness measurements, based on the rationale that CSC is characteristically associated with choroidal thickening [1]. Between January 2023 and October 2024, a total of 361 eyes received intravitreal steroid injections at our institution. Among these patients without concurrent or previous systemic corticosteroid use, 269 eyes were included in the final analysis after excluding 76 eyes with insufficient follow-up and 16 eyes with inadequate choroidal visualization on pre-injection OCT. These 16 eyes showed no CSC features on clinical assessment.

Comprehensive ophthalmologic examination data were extracted from medical records at each visit. Examinations included best-corrected visual acuity (BCVA) measured using Snellen charts and converted to logarithm of the minimum angle of resolution(logMAR) for analysis, slit-lamp biomicroscopy, fundus examination, and spectral-domain optical coherence tomography (SD-OCT) imaging, using Heidelberg Spectralis (Heidelberg Engineering, Heidelberg, Germany). Enhanced depth imaging (EDI) mode was routinely employed to optimise choroidal visualisation. Horizontal line scans (30°×25°, 1024 A-scans) through the foveal centre were obtained with automatic real-time averaging (ART) of at least 25 frames. SFCT measurements were performed manually using calipers software (Heidelberg Eye Explorer software V6.16.8). Measurements of choroidal thickness were obtained beginning at the outer portion of the hyperreflective line corresponding to the retinal pigment epithelium (RPE) and extending to the inner surface of the sclera. In cases where

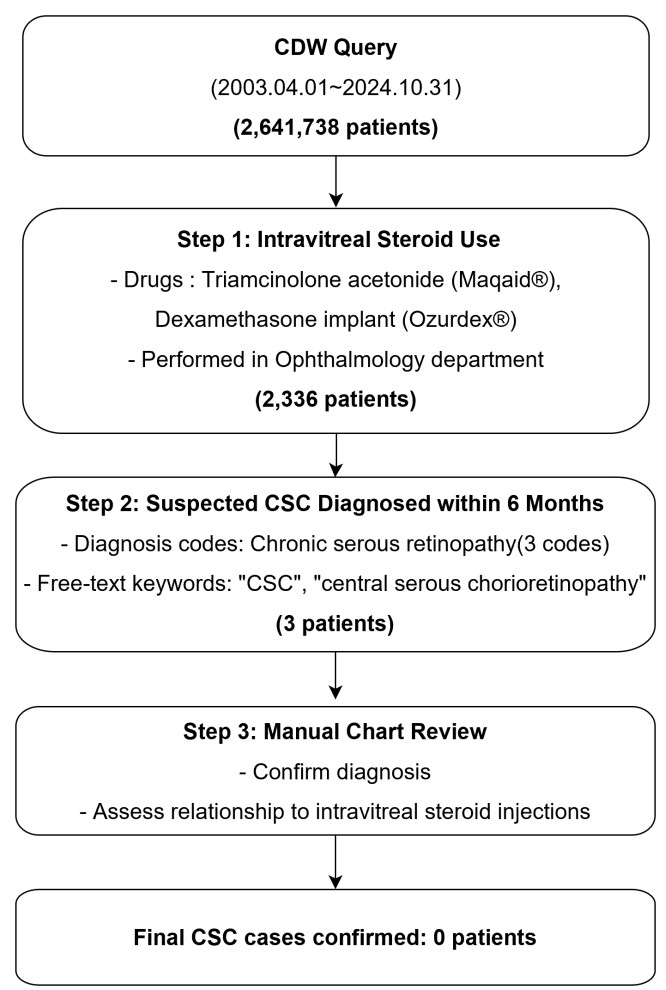

**Fig 1. Flowchart of Clinical Data Warehouse analysis for identifying central serous chorioretinopathy cases following intravitreal steroid injection.** The systematic screening process identified 2,336 patients who received intravitreal triamcinolone acetonide or dexamethasone implant between April 2003 and October 2024. Three suspected cases were identified through diagnosis codes and free-text documentation, all of which were excluded after comprehensive manual review showing no causal association with intravitreal steroid injection. CDW = Clinical Data Warehouse; CSC = central serous chorioretinopathy.

the sclerochoroidal junction was not clearly visible, the eye was excluded from SFCT analysis but included in qualitative assessment for CSC features. All measurements were performed by a single masked ophthalmologist.

For patients receiving multiple injections, the last injection date served as the baseline timepoint. Follow-up data were collected at 1 month, 3 months, and 6 months after the last injection. At each timepoint, BCVA, central macular thickness, and SFCT measurements were measured. Medical records and OCT images were systematically reviewed for any documentation of CSC development, including the presence of subretinal fluid, pigment epithelial detachment, and characteristic choroidal changes.

## Statistical analysis

Descriptive statistics were used to characterize the study population. Categorical variables were presented as frequencies and percentages, while continuous variables were expressed as means with standard deviations. Paired t-tests were used

to compare measurements at different timepoints. The 95% confidence interval for zero observed events was calculated using the rule of three [18]. Statistical significance was set at $p < 0.05$. All analyses were performed using Python 3.11.13 and GraphPad Prism version 10.4.2.

## Results

### Clinical Data Warehouse analysis

The Clinical Data Warehouse (CDW) analysis encompassed a comprehensive 21-year period from April 1, 2003, to October 31, 2024, identifying 2,336 patients who received intravitreal steroid injections at Seoul National University Bundang Hospital (Fig 1). The systematic search included 2 different steroid formulations, with triamcinolone acetonide and dexamethasone implant.

Among these patients, screening for potential CSC diagnoses within 6 months following steroid injection was performed using both structured diagnosis codes containing "Central serous chorioretinopathy" and unstructured free-text clinical documentation containing CSC-related terms ("CSC" and "central serous chorioretinopathy"). This screening identified 3 patients with potential CSC diagnoses.

Each of the 3 suspected cases underwent comprehensive manual chart review by a qualified ophthalmologist, including detailed analysis of clinical notes, optical coherence tomography findings, fluorescein angiography reports, and follow-up treatment plans (S1 Fig).

Case 1 was referred with suspected CSC based on subretinal fluid detected on OCT. However, detailed examination revealed anterior chamber cells and sclerotic vascular changes on funduscopy. Fluorescein angiography demonstrated peripapillary vascular leakage without the characteristic focal leak of CSC. The patient was diagnosed with posterior uveitis and responded well to intravitreal triamcinolone acetonide injection, with complete resolution of subretinal fluid. The initial subretinal fluid was attributed to inflammatory choroidal vascular hyperpermeability rather than CSC.

Case 2 developed subretinal fluid 5 months after uncomplicated cataract surgery (phacoemulsification with posterior chamber intraocular lens implantation) and was initially suspected of having CSC. Intravitreal bevacizumab injection was administered without improvement; instead, intraretinal fluid increased. Following intravitreal triamcinolone acetonide injection, both subretinal and intraretinal fluid completely resolved. The clinical course and response pattern were consistent with pseudophakic cystoid macular oedema with secondary subretinal fluid accumulation rather than primary CSC.

Case 3 had a documented history of CSC that had been treated and resolved several years prior to the study period. Subsequently, the patient developed diabetic macular oedema and was prescribed dexamethasone implant. However, the implant was returned before administration as the treating physician deemed intravitreal bevacizumab more appropriate for this particular case. Only intravitreal bevacizumab was administered; no intravitreal steroid was ever given to this patient. As the dexamethasone implant prescription generated an electronic order in the CDW system, this patient was detected in the initial screening. Since no intravitreal steroid was actually administered, this case was excluded from analysis.

Following comprehensive evaluation, all 3 suspected cases were determined to have no causal association with intravitreal steroid injection. The final confirmed incidence of CSC following intravitreal steroid injection was 0 cases (0/2,336, 0%; 95% CI: 0–0.13%) in this large-scale real-world cohort spanning over two decades.

### Detailed retrospective analysis

The detailed analysis included 269 eyes from patients who received intravitreal steroid injections between January 2023 and October 2024. The cohort was divided into two treatment groups: 114 eyes received Maqaid (triamcinolone acetonide 40 mg) and 154 eyes received Ozurdex (dexamethasone implant 700 mcg). The primary indications for treatment were diabetic macular edema (DME, n = 155, 57.6%), retinal vein occlusion (RVO, n = 71, 26.4%), and uveitis (n = 28, 10.4%).

Baseline demographic and clinical characteristics were well-balanced between treatment groups (Table 1). Mean age was 62.2 ± 10.5 years with equal sex distribution (134 males, 135 females). The prevalence of diabetes mellitus (65.7%) and hypertension (39.6%) showed no significant differences between groups. However, the dexamethasone implant group demonstrated significantly higher baseline central macular thickness (542.1 ± 145.2 μm vs 471.7 ± 148.9 μm, p < 0.001), longer follow-up duration (41.2 ± 30.5 months vs 29.0 ± 31.4 months, p < 0.001), and higher number of steroid injections (5.1 ± 4.4 vs 4.0 ± 4.8, p = 0.0007). Consistent with the CDW analysis findings, no cases of central serous chorioretinopathy were observed throughout the entire follow-up period in the detailed retrospective analysis cohort (0/269 eyes, 0%). This finding reinforced the extremely low incidence rate of CSC following intravitreal steroid injection demonstrated in the larger CDW population.

Contrary to expectations that steroid-induced CSC would be associated with choroidal thickening, both treatment groups demonstrated significant choroidal thinning following intravitreal steroid injection (Table 2 and Fig 2). The most pronounced choroidal thickness reduction occurred at 1 month post-injection in both groups, with the

Table 1. Baseline characteristics of patients who received intravitreal steroid injections. Baseline demographic, clinical, and ocular characteristics of patients included in the detailed retrospective analysis comparing Maqaid (triamcinolone acetonide 40 mg) and Ozurdex (dexamethasone implant 700 mcg) groups. Data are presented as mean ± standard deviation for continuous variables and number (percentage) for categorical variables. Subgroup analyses were performed for the three major indication categories: retinal vein occlusion (RVO), uveitis, and diabetic macular edema (DME). Abbreviations: BCVA = Best-Corrected Visual Acuity; CMT = Central Macular Thickness; DME = Diabetic Macular Edema; DM = Diabetes Mellitus; HTN = Hypertension; RVO = Retinal Vein Occlusion; SFCT = Subfoveal Choroidal Thickness.

| Parameter | Total (n = 269) | Triamcinolone acetonide (n = 114) | Dexamethasone implant (n = 154) | p-value |
|---|---|---|---|---|
| Age, years (range) | 62.2 ± 10.5 (31 - 88) | 63.1 ± 10.3 (33 - 88) | 61.6 ± 10.6 (31 - 85) | 0.237* |
| Sex, Male/Female | 134 / 135 | 56 / 58 | 78 / 76 | 0.902† |
| DM, n (%) | 176 (65.7%) | 73 (64.0%) | 102 (66.7%) | 0.807† |
| HTN, n (%) | 106 (39.6%) | 37 (32.5%) | 69 (45.1%) | 0.055† |
| Indication (RVO/ Uveitis/ DME/ Others) | | | | 0.128† |
| RVO | 71 (26.4%) | 26 (22.8%) | 45 (29.2%) | |
| Uveitis | 28 (10.4%) | 17 (14.9%) | 11 (7.1%) | |
| DME | 155 (57.6%) | 63 (55.3%) | 91 (59.1%) | |
| Others | 15 (5.6%) | 8 (7.0%) | 7 (4.5%) | |
| Baseline BCVA, logMAR | 0.609 ± 0.437 | 0.605 ± 0.397 | 0.616 ± 0.465 | 0.762* |
| RVO subgroup | 0.690 ± 0.497 | 0.676 ± 0.424 | 0.698 ± 0.539 | 0.914* |
| Uveitis subgroup | 0.721 ± 0.434 | 0.702 ± 0.495 | 0.751 ± 0.339 | 0.432* |
| DME subgroup | 0.544 ± 0.410 | 0.532 ± 0.352 | 0.557 ± 0.446 | 0.831* |
| Baseline CMT (μm) | 512.3 ± 150.4 | 471.7 ± 148.9 | 542.1 ± 145.2 | <0.001* |
| RVO subgroup | 498.5 ± 152.4 | 436.6 ± 143.7 | 533.7 ± 147.4 | 0.022* |
| Uveitis subgroup | 600.4 ± 125.0 | 588.1 ± 125.1 | 618.5 ± 128.8 | 0.549* |
| DME subgroup | 502.9 ± 152.0 | 449.0 ± 145.7 | 540.2 ± 146.3 | <0.001* |
| Baseline SFCT (μm) | 231.0 ± 66.7 | 227.1 ± 73.8 | 234.1 ± 61.1 | 0.420* |
| RVO subgroup | 218.4 ± 69.3 | 212.1 ± 80.4 | 222.2 ± 62.5 | 0.586* |
| Uveitis subgroup | 228.8 ± 65.7 | 226.3 ± 76.3 | 232.6 ± 49.2 | 0.637* |
| DME subgroup | 240.0 ± 64.4 | 238.4 ± 69.2 | 241.4 ± 61.4 | 0.787* |
| Follow-up duration, months | 36.0 ± 31.4 | 29.0 ± 31.4 | 41.2 ± 30.5 | <0.001* |
| Number of steroid injections | 4.6 ± 4.6 | 4.0 ± 4.8 | 5.1 ± 4.4 | 0.0007* |

* p-value was calculated by independent T-test or Mann-Whitney U test.

† p-value was calculated by Chi-Square test.

**Table 2. Changes in subfoveal choroidal thickness following intravitreal steroid injections.** Longitudinal changes in subfoveal choroidal thickness (SFCT) measured by optical coherence tomography following intravitreal steroid injections, stratified by treatment type and underlying retinal condition. Data are presented as mean±standard deviation with sample sizes in parentheses. P-values represent statistical significance of changes from last injection to each follow-up timepoint using paired t-tests.

| Subgroup | Time point | Dexamethasone implant, mean±SD (n) | p | Triamcinolone acetonide, mean±SD (n) | p |
|---|---|---|---|---|---|
| **Total** | First injection | 234.1±61.1 (147) | **0.0008** | 227.1±73.8 (111) | **<0.0001** |
| | Last injection | 227.0±65.4 (152) | — | 219.8±78.7 (112) | — |
| | 1 month | 217.9±62.0 (97) | **<0.0001** | 218.6±77.7 (90) | **<0.0001** |
| | 3 months | 219.8±62.7 (115) | **0.0357** | 206.9±76.9 (92) | **<0.0001** |
| | 6 months | 222.8±65.3 (101) | **0.0233** | 215.1±78.9 (79) | **0.032** |
| **RVO** | First injection | 222.2±62.5 (44) | **0.0037** | 212.1±80.4 (26) | **0.042** |
| | Last injection | 208.8±64.7 (44) | — | 200.0±82.7 (26) | — |
| | 1 month | 197.4±63.3 (26) | 0.1527 | 194.0±75.9 (23) | **0.0048** |
| | 3 months | 197.6±68.2 (34) | 0.0661 | 182.3±73.6 (22) | **0.0713** |
| | 6 months | 209.8±75.3 (25) | 0.0819 | 201.5±74.9 (17) | 0.8448 |
| **Uveitis** | First injection | 232.6±49.2 (10) | 0.153 | 226.3±76.3 (15) | 0.2587 |
| | Last injection | 213.4±48.4 (11) | — | 217.5±79.2 (16) | — |
| | 1 month | 198.0±56.6 (7) | 0.6927 | 247.3±79.1 (11) | 0.2659 |
| | 3 months | 217.9±60.2 (10) | 0.7645 | 199.7±64.7 (14) | 0.1979 |
| | 6 months | 195.5±37.7 (6) | 0.9134 | 232.8±86.8 (13) | 0.3834 |
| **DME** | First injection | 241.4±61.4 (86) | **0.0595** | 238.4±69.2 (62) | **0.0002** |
| | Last injection | 237.7±66.1 (90) | — | 233.7±75.4 (62) | — |
| | 1 month | 228.2±61.0 (60) | **0.0001** | 229.7±74.1 (50) | **<0.0001** |
| | 3 months | 232.0±58.9 (67) | 0.3241 | 221.8±79.7 (50) | **<0.0001** |
| | 6 months | 231.1±62.8 (68) | 0.1206 | 220.6±78.2 (45) | **0.004** |

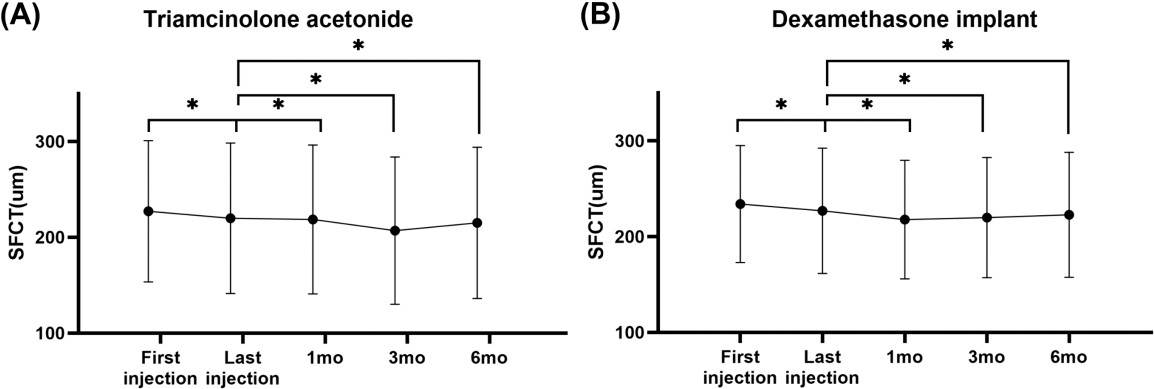

**Fig 2. Serial changes in subfoveal choroidal thickness following intravitreal steroid injections. (A)** Dexamethasone implant group showing progressive choroidal thinning from first injection through 6-month follow-up. **(B)** Triamcinolone acetonide group demonstrating similar choroidal thinning patterns. Data points represent mean values with error bars indicating standard deviation. Statistical significance compared to last injection timepoint: *p<0.05. DME=diabetic macular oedema; RVO=retinal vein occlusion; SFCT=subfoveal choroidal thickness.

dexamethasone implant group showing reduction from 227.0±65.4 µm (last injection) to 217.9±62.0 µm (1 month, p<0.0001) and the triamcinolone acetonide group demonstrating reduction from 219.8±78.7 µm (last injection) to 218.6±77.7 µm (1 month, p<0.0001). The choroidal thinning effect persisted through 6 months in both groups,

with sustained statistical significance compared to last injection measurements. In the total cohort, only 13 patients (4.9%) demonstrated an increase in choroidal thickness of at least 10%, and only 5 patients (1.9%) showed an increase of at least 20% compared to the last injection. Moreover, the dexamethasone implant group showed higher rates of choroidal thickening (11 patients, 7.2% with ≥10% increase; 5 patients, 3.3% with ≥20% increase) compared to the triamcinolone acetonide group (2 patients, 1.8% with ≥10% increase; 0 patients, 0.0% with ≥20% increase). All 5 patients with ≥20% increase in SFCT were diagnosed with RVO or DME. Importantly, none of these patients with choroidal thickening showed subretinal fluid or pigment epithelial detachment on OCT at baseline, last injection, or subsequent follow-up, indicating no features suggestive of CSC (Fig 3). This observation demonstrates that even in the subset of patients who developed choroidal thickening, no CSC developed, further supporting the lack of causal relationship between intravitreal steroids and CSC.

Subgroup analysis revealed distinct patterns across different retinal conditions. The DME subgroup showed the most consistent and sustained choroidal thinning response in both treatment groups, with the triamcinolone acetonide group maintaining significant reduction through 6 months (last injection 233.7 ± 75.4 µm to 220.6 ± 78.2 µm at 6 months, p = 0.004). In contrast, the RVO subgroup demonstrated early choroidal thinning that showed recovery toward last injection values by 6 months, particularly in the triamcinolone acetonide group. The uveitis subgroup exhibited variable responses with no consistent pattern of choroidal thickness changes, reflecting the heterogeneous nature of inflammatory conditions.

Both treatment groups achieved highly significant reduction in macular edema (S1 Table). The most dramatic improvement occurred at 1 month post-injection, with the dexamethasone implant group showing reduction from 492.8 ± 181.2 µm to 326.0 ± 120.0 µm (34% reduction, p < 0.0001) and the triamcinolone acetonide group demonstrating reduction from

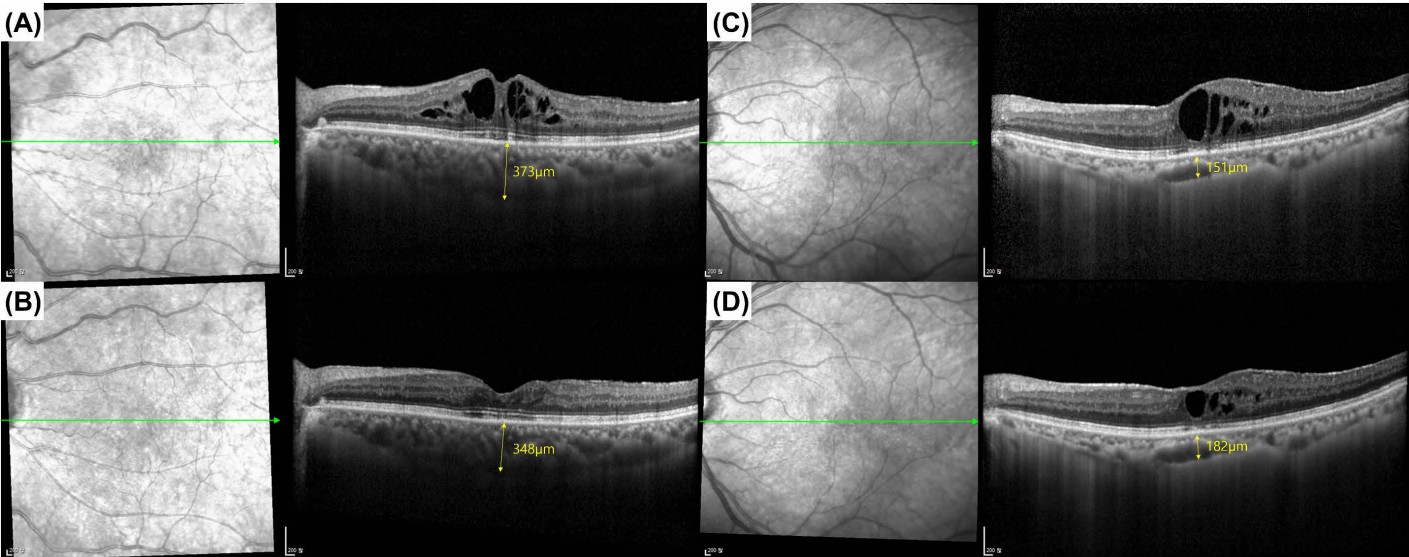

**Fig 3. Representative optical coherence tomography images demonstrating choroidal thickness changes without development of central serous chorioretinopathy. (A, B)** Patient showing choroidal thinning pattern: baseline subfoveal choroidal thickness (SFCT) 373 µm decreased to 348 µm at 3 months post-intravitreal steroid injection. **(C, D)** Patient showing choroidal thickening pattern: baseline SFCT 151 µm increased to 182 µm at 3 months post-injection, representing a 20.5% increase. Note the complete absence of subretinal fluid, pigment epithelial detachment, or other features suggestive of CSC in both cases. Green horizontal line indicates the scan location on the near-infrared reflectance image. CSC = central serous chorioretinopathy; SFCT = subfoveal choroidal thickness.

439.7 ± 146.4 µm to 325.6 ± 101.2 µm (26% reduction, p < 0.0001). These therapeutic effects were maintained through 6 months across all indication subgroups, confirming the efficacy of both steroid formulations for treating macular edema.

## Discussion

This study systematically evaluated the association between intravitreal steroid injections and CSC development through a comprehensive dual approach combining large-scale clinical data surveillance and detailed choroidal thickness validation. Our 21-year CDW analysis revealed a CSC incidence of 0% (0/2,336 patients; 95% CI: 0–0.13%) following intravitreal steroid injection, which was corroborated by our detailed retrospective analysis showing no CSC cases among 269 eyes. Notably, contrary to the choroidal thickening typically associated with CSC, both steroid formulations demonstrated significant choroidal thinning (dexamethasone implant group: 227.0 ± 65.4µm to 217.9 ± 62.0µm, p < 0.0001; triamcinolone acetonide group: 219.8 ± 78.7µm to 218.6 ± 77.7µm, p < 0.0001 at 1 month). These findings suggest that intravitreal steroid injections had no significant association with development of CSC than previously feared, contributing real-world evidence on the safety question for intravitreal steroid injections.

The absence of confirmed CSC cases in our cohort is particularly noteworthy when considered against general population incidence data. Even with conservative estimates of follow-up duration, the observation of zero cases in 2,336 patients suggests a substantially lower incidence than the general population rate of 47 per 100,000 person-years reported by Frederiksen et al [19]. Applying this rate to our cohort over the 6-month surveillance period (2,336 patients × 0.5 years = 1,168 person-years), approximately 0.5–0.6 CSC cases would be expected based on general population incidence alone. While the upper bound of our 95% confidence interval (0.13%, representing a maximum of approximately 3 cases among 2,336 patients) cannot definitively exclude a modest risk increase, several lines of evidence suggest intravitreal steroids do not carry clinically meaningful CSC risk. First, CSC predominantly affects individuals aged between 28 and 68 years, with an average age of 43 [20]. Given our cohort's older age distribution, applying an incidence rate based on the general population may overestimate the expected number of cases, since the incidence rate of CSC decreases after middle age [21]. Second, the observation of zero events despite adequate sample size provides reassurance that any potential risk is not of the magnitude associated with systemic corticosteroid use, which increases CSC risk(OR 4.05 in meta-analysis; adjusted HR 1.81 in a nationwide cohort) [22,23]. Third, our findings are strengthened by the complete absence of CSC even among patients receiving multiple repeated injections, which would be expected to amplify any dose-dependent risk if present. Therefore, while we cannot exclude all possibility of a small risk increase based on the confidence interval alone, we can suggest intravitreal steroid injections do not pose clinically significant CSC risk comparable to systemic corticosteroid administration. This finding is consistent with previous studies from our institution demonstrating that even high-dose systemic corticosteroids have minimal effect on choroidal thickness in most patients, with only one patient (5.6%) developing CSC among 18 patients treated with high-dose systemic steroids, suggesting that steroid-induced CSC may be an idiosyncratic rather than dose-dependent response [24].

The observed choroidal thinning and absence of CSC following intravitreal steroid injection may result from several biological mechanisms. First, local steroid delivery bypasses systemic circulation, minimizing effects on the hypothalamic-pituitary-adrenal axis. Since systemic corticosteroids are thought to induce CSC primarily through mineralocorticoid receptor activation leading to fluid retention and increased vascular permeability, intravitreal injection bypasses these systemic pathways [1,25]. Second, the choroidal thinning observed likely represents anti-inflammatory effects causing choroidal vessel constriction and reduced vascular permeability. This mechanism aligns with the therapeutic response seen in inflammatory retinal conditions, where choroidal thickening secondary to inflammation resolves following anti-inflammatory treatment [26,27]. Moreover, regarding choroidal thickness changes, multiple studies have reported choroidal thinning following anti-VEGF therapy or steroid treatment in various retinal conditions [27–30]. This is consistent with our OCT findings across all indication subgroups. While our study lacks a concurrent control group, making it impossible to definitively distinguish steroid effects from natural disease progression, the key finding for our research question is not

whether choroidal thinning represents steroid effect versus natural history(which typically shows stable choroidal thickness when untreated) [27], but rather that choroidal thickening (the hallmark feature of CSC) was not observed following intravitreal steroid injection. The absence of choroidal thickening and CSC development across all disease subgroups provides supporting evidence against a causal relationship between intravitreal steroid administration and CSC, which was the primary objective of our investigation.

Several studies have reported findings that contrast with our results. Imasawa et al. described CSC development following vitrectomy with intravitreal triamcinolone acetonide for diabetic macular edema [11], while Baumal reported CSC associated with periocular corticosteroid injection [12]. More recently, Georgalas et al. and Noh et al. reported CSC cases following intravitreal dexamethasone implants [13,14]. Additionally, Ersoz et al. reported a case of sequential development of pachychoroid pigment epitheliopathy and transformation to CSC after repeated intravitreal dexamethasone implantations, suggesting a potential cumulative effect of consecutive steroid administrations [31]. These discrepancies may be explained by the case reports involving different patient populations, steroid formulations, and concurrent procedures that could confound CSC development. Furthermore, the possibility of idiosyncratic reactions in genetically predisposed individuals cannot be excluded. Moreover, our findings contrast with broader injection-related CSC risks reported in meta-analyses. Ge et al.'s 2020 meta-analysis found an odds ratio of 1.66 for "injection" as a general category [22], but this encompassed intravenous, intramuscular, and topical injections, making it difficult to extrapolate specifically to intravitreal administration. Our targeted analysis of the intravitreal route provides more precise risk estimates for this specific delivery method, suggesting that local ocular delivery may have a distinctly different risk profile compared to systemic or other parenteral routes.

Regarding our study population, while our cohort comprised patients with diabetic macular edema, retinal vein occlusion, and uveitis rather than the general population, this represents the clinically relevant population of interest. The clinical question is not whether intravitreal steroids increase CSC risk in healthy individuals, but rather whether they increase CSC risk in patients who require intravitreal steroid treatment for retinal diseases. Our study population therefore appropriately reflects the target population for risk assessment and clinical decision-making. The mean age of 62.2 years in our cohort, whilst older than the typical CSC demographic (middle-aged men, 30–50 years), reflects the age distribution of patients requiring intravitreal steroid therapy for the studied indications.

Moreover, our results align with the safety profiles reported in pivotal clinical trials for intravitreal corticosteroids. The FAME studies, GENEVA studies, and DRCR.net trial did not report CSC as a safety concern [32–36]. This concordance between our real-world evidence and controlled trial data, combined with the mechanistic finding of choroidal thinning rather than thickening, suggests that the association reported in isolated case reports may represent idiosyncratic reactions in genetically predisposed individuals rather than a class effect. From a clinical decision-making perspective, routine avoidance of intravitreal corticosteroids based solely on theoretical CSC concerns may not be warranted in the general patient population. However, heightened monitoring remains prudent in patients with pre-existing pachychoroid features, documented CSC history, or other known risk factors such as systemic corticosteroid use. Future prospective studies are needed to examine CSC development in these high-risk subgroups.

The complementary dual approach employed in this study addressed limitations inherent in each method alone. The CDW analysis provided comprehensive surveillance across all patients over an extended period but relied on clinical documentation. However, given that all intravitreal steroid injections at our institution are performed by retinal specialists who directly follow up these patients with routine OCT examinations, any development of subretinal fluid or CSC-like features would be documented during regular post-injection examinations. The probability of a retinal specialist identifying subretinal fluid on OCT but failing to document it in either diagnosis codes or clinical notes is extremely low. The detailed OCT analysis offered objective validation through choroidal thickness measurements, based on the rationale that CSC characteristically shows choroidal thickening. Therefore, if intravitreal steroids cause CSC, choroidal thickening should be observed. The finding of choroidal thinning, and the absence of CSC even in the small subset who developed choroidal thickening, provides supporting evidence against a causal relationship.

Several limitations should be acknowledged. The retrospective design introduces potential selection bias and limits control over confounding variables. As a single-center study, findings may not be generalizable to different populations or practice patterns. OCT measurement limitations include potential artifacts from media opacities and the manual measurement approach. The study population was ethnically homogeneous, limiting applicability to diverse populations. Additionally, while our CDW analysis spanned 21 years, individual patient follow-up durations were variable, and patients with shorter follow-up periods may theoretically have had CSC events that were not captured. Future multicenter and prospective randomized controlled trials for intravitreal steroid injections are needed to confirm generalizability. Moreover, our study focused exclusively on intravitreal steroid injections and did not include subtenon steroid administrations, which require further investigation due to different delivery mechanisms. Furthermore, while we used subfoveal choroidal thickness as a surrogate marker for CSC development, this approach may not capture cases where CSC presents with non-subfoveal subretinal fluid and predominantly extrafoveal choroidal thickening. Despite these limitations, the study has several strengths. The comprehensive 21-year observation period represents the largest cohort studied for this indication. Manual chart review by qualified ophthalmologists ensured diagnostic accuracy and eliminated false-positive cases. Moreover, dual approach combining population-level surveillance with detailed imaging analysis overcame limitations of previous case report-based evidence. The subgroup analysis according to indications may enhance the generalizability of findings within the context of routine clinical practice. The demonstration that even patients developing choroidal thickening did not develop CSC provides evidence against a causal relationship.

In conclusion, intravitreal steroid injections showed no confirmed cases of CSC in our large real-world cohort spanning over two decades, with a 95% confidence interval (0–0.13%) suggesting substantially lower risk than general population incidence rates and markedly lower than risks associated with systemic corticosteroid use. The choroidal thinning observed following both dexamethasone and triamcinolone injections contradicts the choroidal thickening typically associated with CSC and suggests distinct pathophysiological mechanisms. These findings provide valuable evidence to guide clinical decision-making and patient counselling regarding CSC risk after intravitreal steroid injections.

## Supporting information

**S1 Fig. Detailed optical coherence tomography and clinical findings of three suspected central serous chorioretinopathy cases identified in Clinical Data Warehouse screening.** (A) Case 1: Posterior uveitis initially presenting with subretinal fluid that was suspected as CSC. Fundus examination revealed anterior chamber cells and sclerotic vascular changes. Fluorescein angiography demonstrated peripapillary vascular leakage without characteristic focal leak of CSC. Intravitreal triamcinolone acetonide injection was done. (B) Case 2: Pseudophakic cystoid macular edema with secondary subretinal fluid developing 5 months after uncomplicated cataract surgery. Initial treatment with intravitreal bevacizumab showed no improvement with increasing intraretinal fluid. Complete resolution of both subretinal and intraretinal fluid occurred after intravitreal triamcinolone acetonide injection, consistent with pseudophakic cystoid macular oedema rather than primary CSC. Top row shows baseline presentation, middle row shows post-bevacizumab worsening, and bottom row shows resolution after triamcinolone injection. CSC = central serous chorioretinopathy.
(DOCX)

**S1 Table. Changes in central macular thickness following intravitreal steroid injections.** Longitudinal changes in central macular thickness (CMT) measured by optical coherence tomography following intravitreal steroid injections, stratified by treatment type and underlying retinal condition. Data are presented as mean ± standard deviation with sample sizes in parentheses. P-values represent statistical significance of changes from last injection to each follow-up timepoint using paired t-tests. Abbreviations: CMT = Central Macular Thickness; DME = Diabetic Macular Edema; RVO = Retinal Vein Occlusion; SD = Standard Deviation.
(DOCX)

## Author contributions

**Conceptualization:** Jae Ryong Song, Se Joon Woo.

**Data curation:** Jae Ryong Song.

**Formal analysis:** Jae Ryong Song, Se Joon Woo.

**Investigation:** Jae Ryong Song.

**Supervision:** Se Joon Woo.

**Writing – original draft:** Jae Ryong Song.

**Writing – review & editing:** Jae Ryong Song, Se Joon Woo.

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
