## [Decision Letter · Decision Letter 0]

3 Dec 2025

Dear Dr. Woo,

Thank you for submitting your manuscript to PLOS ONE. After careful consideration, we feel that it has merit but does not fully meet PLOS ONE’s publication criteria as it currently stands. Therefore, we invite you to submit a revised version of the manuscript that addresses the points raised during the review process.

We look forward to receiving your revised manuscript.

Kind regards,

Alon Harris

Academic Editor

PLOS ONE

Journal Requirements:

2. Please expand the acronym “MSIT” (as indicated in your financial disclosure) so that it states the name of your funders in full.

3. In the online submission form you indicate that your data is not available for proprietary reasons and have provided a contact point for accessing this data. Please note that your current contact point is a co-author on this manuscript. According to our Data Policy, the contact point must not be an author on the manuscript and must be an institutional contact, ideally not an individual. Please revise your data statement to a non-author institutional point of contact, such as a data access or ethics committee, and send this to us via return email. Please also include contact information for the third party organization, and please include the full citation of where the data can be found.

Reviewers' comments:

Reviewer's Responses to Questions

**Comments to the Author**

1. Is the manuscript technically sound, and do the data support the conclusions?

Reviewer #1: Yes

Reviewer #2: Yes

2. Has the statistical analysis been performed appropriately and rigorously?

Reviewer #1: I Don't Know

Reviewer #2: Yes

3. Have the authors made all data underlying the findings in their manuscript fully available?

Reviewer #1: No

Reviewer #2: Yes

4. Is the manuscript presented in an intelligible fashion and written in standard English?

Reviewer #1: Yes

Reviewer #2: Yes

Reviewer #1: 1. This is a clinically important study addressing a mechanistically intriguing and very rare event. Prior evidence is mostly isolated case reports; a 21-year, zero-event cohort is highly reassuring. It would help if the authors more explicitly compare their upper 95% CI with the expected background incidence (e.g., estimated expected CSC cases in this population), to better frame the clinical impact.

2. OCT availability and choroidal thickness methodology

For the CDW cohort, please clarify whether OCT data were directly used or whether CSC screening relied only on codes and free text. For the detailed cohort, specify:

Whether EDI-OCT was used for all included eyes.

What proportion of all intravitreal steroid–treated eyes in 2023–2024 had analyzable OCT and why others were excluded.

This is important for understanding selection bias and how broadly imaging findings apply.

3. Definition and adjudication of CSC

Please provide a clear operational definition of CSC (clinical + OCT ± angiography criteria) and describe the adjudication process (number of reviewers, how discrepancies were handled). This will reassure readers that

4. Systemic/other steroid exposure and prior CSC

Since the main concern is “steroid-induced” CSC, more detail on other steroid exposures would strengthen the

Reviewer #2: The authors address an important clinical question - do intravitreal corticosteroid injections/implants increase the risk of developing CSC - and provide a clear, well-supported answer.

Comments:

1. The abstract could benefit from being re-organized.

2. What was the rationale for limiting the search to 6 months after intravitreal injection?

3. Clarify in the methods whether patients received both “triamcinolone acetonide and dexamethasone implant” (as stated at the moment) or one of the two.

4. Loss to follow-up: is all post-injection follow-up performed in-house? How many patients are missing data post-injection?

5. The use of SFCT as a surrogate for CSC is reasonable, however - CSC often presents in a non-subfoveal location, which should be acknowledged.

6. There is no control group of similar patients receiving no or different treatment - could the observed thinning reflect the natural history of the underlying pathology rather than steroid effect?

7. In Table 2, parentheses around the n values are misplaced.

8. Please report the range of ages (not just mean ± SD) to clarify whether patients in the typical CSC age group were included, and if not - address.

Overall, the study is of high clinical relevance. Addressing these points would strengthen the manuscript and improve interpretability.

**Do you want your identity to be public for this peer review?** For information about this choice, including consent withdrawal, please see our For information about this choice, including consent withdrawal, please see our Privacy Policy .

Reviewer #1: No

Reviewer #2: No

---

## [Author Response · Author response to Decision Letter 1]

26 Dec 2025

December 24, 2025

Manuscript ID: PONE-D-25-57894

Title: Intravitreal injections of corticosteroid and the risk of central serous chorioretinopathy

Dear Editor and Reviewers,

We sincerely appreciate the thoughtful and constructive feedback from both reviewers. Your comments have significantly improved the quality and clarity of our manuscript. All comments from the reviewers were very thoughtful and have helped us to improve our manuscript. We have carefully checked our manuscript and made appropriate revisions based on the reviewers’ suggestions.

A point-by-point response to the editor’s comments is presented below.

Response to Reviewer #1

Reviewer Comment 1:

"This is a clinically important study addressing a mechanistically intriguing and very rare event. Prior evidence is mostly isolated case reports; a 21-year, zero-event cohort is highly reassuring. It would help if the authors more explicitly compare their upper 95% CI with the expected background incidence (e.g., estimated expected CSC cases in this population), to better frame the clinical impact."

Response: We thank the reviewer for this excellent suggestion. We have added a more explicit quantitative comparison in the Discussion section, while acknowledging the statistical limitations of zero-event observations.

Lines 298-314:

“Applying this rate to our cohort over the 6-month surveillance period (2,336 patients × 0.5 years = 1,168 person-years), approximately 0.5-0.6 CSC cases would be expected based on general population incidence alone. While the upper bound of our 95% confidence interval (0.13%, representing a maximum of approximately 3 cases among 2,336 patients) cannot definitively exclude a modest risk increase, several lines of evidence suggest intravitreal steroids do not carry clinically meaningful CSC risk. First, CSC predominantly affects individuals aged between 28 and 68 years, with an average age of 43.[18] Given our cohort’s older age distribution, applying an incidence rate based on the general population may overestimate the expected number of cases, since the incidence rate of CSC decreases after middle age.[19] Second, the observation of zero events despite adequate sample size provides reassurance that any potential risk is not of the magnitude associated with systemic corticosteroid use, which increases CSC risk(OR 4.05 in meta-analysis; adjusted HR 1.81 in a nationwide cohort).[20,21] Third, our findings are strengthened by the complete absence of CSC even among patients receiving multiple repeated injections, which would be expected to amplify any dose-dependent risk if present. Therefore, while we cannot exclude all possibility of a small risk increase based on the confidence interval alone, we can suggest intravitreal steroid injections do not pose clinically significant CSC risk comparable to systemic corticosteroid administration."

20. Gäckle HC, Lang GE, Freissler KA, Lang GK. [Central serous chorioretinopathy. Clinical, fluorescein angiography and demographic aspects]. Ophthalmol Z Dtsch Ophthalmol Ges. 1998;95: 529–533. doi:10.1007/s003470050310

21. Tsai D-C, Chen S-J, Huang C-C, Chou P, Chung C-M, Huang P-H, et al. Epidemiology of Idiopathic Central Serous Chorioretinopathy in Taiwan, 2001–2006: A Population-based Study. PLOS ONE. 2013;8: e66858. doi:10.1371/journal.pone.0066858

22. Ge G, Zhang Y, Zhang Y, Xu Z, Zhang M. Corticosteroids usage and central serous chorioretinopathy: a meta-analysis. Graefes Arch Clin Exp Ophthalmol. 2020;258: 71–77. doi:10.1007/s00417-019-04486-w

23. Rim TH, Kim HS, Kwak J, Lee JS, Kim DW, Kim SS. Association of Corticosteroid Use With Incidence of Central Serous Chorioretinopathy in South Korea. JAMA Ophthalmol. 2018;136: 1164–1169. doi:10.1001/jamaophthalmol.2018.3293

Reviewer Comment 2:

"OCT availability and choroidal thickness methodology. For the CDW cohort, please clarify whether OCT data were directly used or whether CSC screening relied only on codes and free text. For the detailed cohort, specify: Whether EDI-OCT was used for all included eyes. What proportion of all intravitreal steroid–treated eyes in 2023–2024 had analyzable OCT and why others were excluded. This is important for understanding selection bias and how broadly imaging findings apply."

Response:

We appreciate this important clarification request. We have revised the Methods section to address these concerns comprehensively.

Lines 103-106:

"Only the three suspected cases meeting both injection and diagnosis codes or free-text clinical notes underwent comprehensive manual review by qualified ophthalmologists (JRS, SJW) to determine the temporal and causal relationship between steroid injection and CSC development.

Regarding EDI-OCT usage, this has already been specified in the Methods section (Lines 135-136): "Enhanced depth imaging (EDI) mode was routinely employed to optimise choroidal visualisation."

We have now added information about the proportion of analyzable OCT data:

Lines 127-131

" Between January 2023 and October 2024, a total of 361 eyes received intravitreal steroid injections at our institution. Among these patients without concurrent or previous systemic corticosteroid use, 269 eyes were included in the final analysis after excluding 76 eyes with insufficient follow-up and 16 eyes with inadequate choroidal visualization on pre-injection OCT. These 16 eyes showed no CSC features on clinical assessment.”

Reviewer Comment 3:

"Definition and adjudication of CSC. Please provide a clear operational definition of CSC (clinical + OCT ± angiography criteria) and describe the adjudication process (number of reviewers, how discrepancies were handled). This will reassure readers that..."

Response:

We agree that a clear operational definition and adjudication process is essential. We have added the following to the Methods section:

Lines 106-110:

"CSC was defined as subretinal fluid on optical coherence tomography in the macular region without alternative explanation. When available, fluorescein angiography findings such as focal leakage points provided additional diagnostic confirmation. Moreover, temporal relationships with steroid injections were evaluated. Discrepancies were resolved through consensus discussion.”

Reviewer Comment 4:

"Systemic/other steroid exposure and prior CSC. Since the main concern is "steroid-induced" CSC, more detail on other steroid exposures would strengthen the..."

Response:

We thank the reviewer for highlighting this important methodological detail. We have clarified our exclusion criteria to the Methods section:

Lines 127-131

" Between January 2023 and October 2024, a total of 361 eyes received intravitreal steroid injections at our institution. Among these patients without concurrent or previous systemic corticosteroid use, 269 eyes were included in the final analysis after excluding 76 eyes with insufficient follow-up and 16 eyes with inadequate choroidal visualization on pre-injection OCT. These 16 eyes showed no CSC features on clinical assessment.”

Response to Reviewer #2

Reviewer Comment 1:

"The abstract could benefit from being re-organized."

Response:

We have comprehensively reorganized the abstract to improve flow and clarity. The revised abstract now follows a more logical structure:

Line 21-37:

“Central serous chorioretinopathy (CSC) is strongly associated with systemic corticosteroid use, but the risk following intravitreal steroid injections remains unclear, with limited evidence from isolated case reports. This study evaluated the association between intravitreal steroid injections and CSC development using a dual-approach design with Clinical Data Warehouse (CDW) analysis and choroidal thickness validation. We performed CDW analysis of 2,336 patients receiving intravitreal triamcinolone acetonide or dexamethasone implant over 21 years (April 2003–October 2024), screening for CSC within 6 months post-injection. Additionally, we conducted detailed subfoveal choroidal thickness analysis in 269 eyes (January 2023–October 2024) at baseline, 1, 3, and 6 months post-injection using optical coherence tomography. No confirmed CSC cases occurred in either cohort (0/2,336 patients, 95% CI: 0–0.13%; 0/269 eyes). Contrary to the choroidal thickening characteristic of CSC, both steroid formulations produced significant choroidal thinning at 1 month: dexamethasone implant (227.0±65.4μm to 217.9±62.0μm, p<0.0001) and triamcinolone acetonide (219.8±78.7μm to 218.6±77.7μm, p<0.0001). Only 4.9% of eyes showed ≥10% choroidal thickening, and none developed CSC features. Both formulations achieved significant macular edema reduction through 6 months. Intravitreal corticosteroids showed no confirmed CSC cases over two decades with choroidal thinning rather than the thickening characteristic of CSC, suggesting distinct pathophysiological mechanisms and a favorable safety profile regarding CSC development.”

Reviewer Comment 2:

"What was the rationale for limiting the search to 6 months after intravitreal injection?"

Response:

We thank the reviewer for this important question. We have added justification for the 6-month timeframe based on existing case report literature. The following has been added to the Methods section:

Lines 97-99:

"6 months post-injection based on temporal patterns reported in previous case reports[11–14] and pharmacokinetic studies of dexamethasone implants showing sustained drug release for up to 6 months with negligible residual intraocular steroid thereafter.[16,17]”

11. Imasawa M, Ohshiro T, Gotoh T, Imai M, Iijima H. Central serous chorioretinopathy following vitrectomy with intravitreal triamcinolone acetonide for diabetic macular oedema. Acta Ophthalmol Scand. 2005;83: 132–133. doi:10.1111/j.1600-0420.2005.00379.x

12. Baumal CR. Central Serous Chorioretinopathy Associated With Periocular CorticosteroidInjection Treatment for HLA-B27–Associated Iritis. Arch Ophthalmol. 2004;122: 926. doi:10.1001/archopht.122.6.926

13. Georgalas I, Petrou P, Pagoulatos D, Papaconstantinou D, Tservakis I. Central serous chorioretinopathy in the fellow eye as a complication of intravitreal dexamethasone implant for the treatment of Irvine‐Gass syndrome. Clin Exp Optom. 2016;99: 601–603. doi:10.1111/cxo.12420

14. Noh GM, Nam KY, Lee SU, Lee SJ. Central Serous Chorioretinopathy Following Intravitreal Dexamethasone Implant. Korean J Ophthalmol. 2019;33: 392. doi:10.3341/kjo.2018.0115

16. Chang-Lin J-E, Burke JA, Peng Q, Lin T, Orilla WC, Ghosn CR, et al. Pharmacokinetics of a Sustained-Release Dexamethasone Intravitreal Implant in Vitrectomized and Nonvitrectomized Eyes. Invest Ophthalmol Vis Sci. 2011;52: 4605–4609. doi:10.1167/iovs.10-6387

17. Chang-Lin J-E, Attar M, Acheampong AA, Robinson MR, Whitcup SM, Kuppermann BD, et al. Pharmacokinetics and pharmacodynamics of a sustained-release dexamethasone intravitreal implant. Invest Ophthalmol Vis Sci. 2011;52: 80–86. doi:10.1167/iovs.10-5285

Reviewer Comment 3:

"Clarify in the methods whether patients received both 'triamcinolone acetonide and dexamethasone implant' (as stated at the moment) or one of the two."

Response:

We apologize for this ambiguity. We have clarified that patients received one of the two steroid formulations, not both concurrently. The following revisions have been made:

Lines 122-124:

"A detailed retrospective analysis was conducted on patients who received intravitreal steroid injections with either triamcinolone acetonide 40 mg (Maqaid, Hanmi Pharm., Seoul, Korea) or dexamethasone implant 700 mcg (Ozurdex, Korea AbbVie Ltd., Seoul, Korea)) between January 2023 and October 2024. "

Reviewer Comment 4:

"Loss to follow-up: is all post-injection follow-up performed in-house? How many patients are missing data post-injection?"

Response: We have now added information about the proportion of analyzed OCT data:

Lines 127-131:

" Between January 2023 and October 2024, a total of 361 eyes received intravitreal steroid injections at our institution. Among these patients without concurrent or previous systemic corticosteroid use, 269 eyes were included in the final analysis after excluding 76 eyes with insufficient follow-up and 16 eyes with inadequate choroidal visualization on pre-injection OCT. These 16 eyes showed no CSC features on clinical assessment."

Reviewer Comment 5:

"The use of SFCT as a surrogate for CSC is reasonable, however - CSC often presents in a non-subfoveal location, which should be acknowledged."

Response:

This is an excellent point that we have added to the limitations

Line 383-385:

“Furthermore, while we used subfoveal choroidal thickness as a surrogate marker for CSC development, this approach may not capture cases where CSC presents with non-subfoveal subretinal fluid and predominantly extrafoveal choroidal thickening."

Reviewer Comment 6:

"There is no control group of similar patients receiving no or different treatment - could the observed thinning reflect the natural history of the underlying pathology rather than steroid effect?"

Response:

We appreciate this thoughtful consideration of alternative explanations. We have enhanced our Discussion to address this important point:

Lines 329-336:

“While our study lacks a concurrent control group, making it impossible to definitively distinguish steroid effects from natural disease progression, the key finding for our research question is not whether choroidal thinning represents steroid effect versus natural history(which typically shows stable choroidal thickness when untreated)[26], but rather that choroidal thickening (the hallmark feature of CSC) was not observed following intravitreal steroid injection. The absence of choroidal thickening and CSC development across all disease subgroups provides supporting evidence against a causal relationship between intravitreal steroid administration and CSC, which was the primary objective of our investigation.”

27. Yiu G, Manjunath V, Chiu SJ, Farsiu S, Mahmoud TH. Effect of Anti–Vascular Endothelial Growth Factor Therapy on Choroidal Thickness in Diabetic Macular Edema. Am J Ophthalmol. 2014;158: 745-751.e2. doi:10.1016/j.ajo.2014.06.006

Reviewer Comment 7:

"In Table 2, parentheses around the n values are misplaced."

Response:

We have carefully reviewed Table 2 and corrected the parentheses placement.

Reviewer Comment 8:

"Please report the range of ages (not just mean ± SD) to clarify whether patients in the typical CSC age group were included, and if not - address."

Response: Thanks for your comments. We added the range of ages at Table 1.

Thank you for your detailed feedback and consideration of our submitted paper. Your feedback has helped us address important issues and significantly improve our manuscript. We believe these comprehensive revisions have enhanced the scientific rigor, clarity, and clinical relevance of our work. We hope our responses have adequately addressed your comments and that our revised manuscript is now suitable for publication in PLOS ONE.

Sincerely,

Se Joon Woo, MD, PhD

Department of Ophthalmology

Seoul National University Bundang Hospital

82, Gumi-ro 173beon-gil, Bundang-gu, Seongnam, Gyeonggi-do, South Korea, 13620

Tel: +82-31-787-7377

Email: sejoon1@snu.ac.kr

---

## [Decision Letter · Decision Letter 1]

20 Jan 2026

Intravitreal injections of corticosteroid and the risk of central serous chorioretinopathy

PLOS One

Dear Dr. Woo,

Thank you for submitting your manuscript to PLOS ONE. After careful consideration, we feel that it has merit but does not fully meet PLOS ONE’s publication criteria as it currently stands. Therefore, we invite you to submit a revised version of the manuscript that addresses the points raised during the review process.

We look forward to receiving your revised manuscript.

Kind regards,

Alon Harris

Academic Editor

PLOS One

**Journal Requirements:**

Reviewers' comments:

Reviewer's Responses to Questions

**Comments to the Author**

Reviewer #1: (No Response)

Reviewer #2: All comments have been addressed

2. Is the manuscript technically sound, and do the data support the conclusions?

Reviewer #1: Yes

Reviewer #2: Yes

3. Has the statistical analysis been performed appropriately and rigorously?

Reviewer #1: I Don't Know

Reviewer #2: Yes

4. Have the authors made all data underlying the findings in their manuscript fully available?

Reviewer #1: Yes

Reviewer #2: Yes

5. Is the manuscript presented in an intelligible fashion and written in standard English?

Reviewer #1: Yes

Reviewer #2: Yes

Reviewer #1: I would like to thank the authors for addressing an important yet clinically vague topic and for carefully revising the manuscript

1. Free-text screening methodology in the CDW cohort

Given that OCT images were not systematically reviewed for the majority of the CDW cohort, the accuracy of CSC detection relies heavily on diagnosis codes and free-text searches. To improve transparency and reproducibility, the Methods section would benefit from a more detailed description of the free-text search strategy. Specifically, listing the exact search terms (and whether variants, abbreviations, or non-English terms were included), similar to how search strings are reported in systematic reviews, would strengthen confidence in the completeness of case ascertainment.

2.

The relationship between Intra virtual corticosteroid exposure and central serous chorioretinopathy remains a source of uncertainty in daily practice. Many clinicians continue to avoid intravitreal steroids in patients with pachychoroid spectrum disease on their differential, despite the evidence being largely limited to isolated case reports.

The Discussion could be further strengthened by explicitly framing how these findings together with the fact that CSCR wasn't raised as a safety issue in the pivotal trials approving these preparations/implant should influence clinical decision-making. Otherwise, suggest which additional work is required to make such a conclusion

Reviewer #2: The authors have addressed all comments satisfactorily.

The revisions clarify methodology, justify the 6-month follow-up window, transparently report exclusions and follow-up, and appropriately acknowledge key limitations, including the use of SFCT and the lack of a control group.

Overall, the changes improve clarity and interpretability, and the manuscript is suitable for publication

**Do you want your identity to be public for this peer review?** For information about this choice, including consent withdrawal, please see our For information about this choice, including consent withdrawal, please see our Privacy Policy .

Reviewer #1: No

Reviewer #2: No

---

## [Author Response · Author response to Decision Letter 2]

21 Jan 2026

January 21, 2026

Manuscript ID: PONE-D-25-57894

Title: Intravitreal injections of corticosteroid and the risk of central serous chorioretinopathy

Dear Editor and Reviewers,

We sincerely thank the reviewer for the continued careful evaluation of our manuscript and for the constructive suggestions that further improve its transparency and clinical relevance. We have addressed all comments in detail as outlined below.

Response to Reviewer #1

Reviewer Comment 1: I would like to thank the authors for addressing an important yet clinically vague topic and for carefully revising the manuscript

"Free-text screening methodology in the CDW cohort

Given that OCT images were not systematically reviewed for the majority of the CDW cohort, the accuracy of CSC detection relies heavily on diagnosis codes and free-text searches. To improve transparency and reproducibility, the Methods section would benefit from a more detailed description of the free-text search strategy. Specifically, listing the exact search terms (and whether variants, abbreviations, or non-English terms were included), similar to how search strings are reported in systematic reviews, would strengthen confidence in the completeness of case ascertainment."

Response: We appreciate this valuable suggestion to enhance the transparency and reproducibility of our search methodology. We have revised the Methods section to provide comprehensive details of our free-text search strategy, including the exact search terms and inclusion of abbreviations and Korean equivalents.

Line 94-98

These patients were then screened for CSC diagnosis following steroid injection using both structured diagnosis codes containing 'Central serous chorioretinopathy' and unstructured free-text clinical notes containing CSC-related terms ('CSC', ‘CSCR’, 'central serous chorioretinopathy' and Korean equivalents) within 6 months post-injection based on temporal patterns reported in previous case reports

“The relationship between Intra virtual corticosteroid exposure and central serous chorioretinopathy remains a source of uncertainty in daily practice. Many clinicians continue to avoid intravitreal steroids in patients with pachychoroid spectrum disease on their differential, despite the evidence being largely limited to isolated case reports.

The Discussion could be further strengthened by explicitly framing how these findings together with the fact that CSCR wasn't raised as a safety issue in the pivotal trials approving these preparations/implant should influence clinical decision-making. Otherwise, suggest which additional work is required to make such a conclusion”

Response: We thank the reviewer for this insightful comment regarding the clinical implications of our findings. We have added a new paragraph to the Discussion section that addresses this concern.

Line 361-371:

“Moreover, our results align with the safety profiles reported in pivotal clinical trials for intravitreal corticosteroids. The FAME studies, GENEVA studies, and DRCR.net trial did not report CSC as a safety concern.[32–36] This concordance between our real-world evidence and controlled trial data, combined with the mechanistic finding of choroidal thinning rather than thickening, suggests that the association reported in isolated case reports may represent idiosyncratic reactions in genetically predisposed individuals rather than a class effect. From a clinical decision-making perspective, routine avoidance of intravitreal corticosteroids based solely on theoretical CSC concerns may not be warranted in the general patient population. However, heightened monitoring remains prudent in patients with pre-existing pachychoroid features, documented CSC history, or other known risk factors such as systemic corticosteroid use. Future prospective studies are needed to examine CSC development in these high-risk subgroups.”

32. Campochiaro PA, Brown DM, Pearson A, Chen S, Boyer D, Ruiz-Moreno J, et al. Sustained delivery fluocinolone acetonide vitreous inserts provide benefit for at least 3 years in patients with diabetic macular edema. Ophthalmology. 2012;119: 2125–2132. doi:10.1016/j.ophtha.2012.04.030

33. Campochiaro PA, Brown DM, Pearson A, Ciulla T, Boyer D, Holz FG, et al. Long-term benefit of sustained-delivery fluocinolone acetonide vitreous inserts for diabetic macular edema. Ophthalmology. 2011;118: 626-635.e2. doi:10.1016/j.ophtha.2010.12.028

34. Haller JA, Bandello F, Belfort R, Blumenkranz MS, Gillies M, Heier J, et al. Randomized, sham-controlled trial of dexamethasone intravitreal implant in patients with macular edema due to retinal vein occlusion. Ophthalmology. 2010;117: 1134-1146.e3. doi:10.1016/j.ophtha.2010.03.032

35. Haller JA, Bandello F, Belfort R, Blumenkranz MS, Gillies M, Heier J, et al. Dexamethasone intravitreal implant in patients with macular edema related to branch or central retinal vein occlusion twelve-month study results. Ophthalmology. 2011;118: 2453–2460. doi:10.1016/j.ophtha.2011.05.014

36. Diabetic Retinopathy Clinical Research Network. A randomized trial comparing intravitreal triamcinolone acetonide and focal/grid photocoagulation for diabetic macular edema. Ophthalmology. 2008;115: 1447–1449, 1449.e1–10. doi:10.1016/j.ophtha.2008.06.015

Response to Reviewer #2

“The authors have addressed all comments satisfactorily. The revisions clarify methodology, justify the 6-month follow-up window, transparently report exclusions and follow-up, and appropriately acknowledge key limitations, including the use of SFCT and the lack of a control group. Overall, the changes improve clarity and interpretability, and the manuscript is suitable for publication.”

Response:

We thank Reviewer #2 for their positive evaluation and for confirming that all previous comments have been satisfactorily addressed.

Thank you for your detailed feedback and consideration of our submitted paper. Your feedback has helped us address important issues and significantly improve our manuscript. We believe these comprehensive revisions have enhanced the scientific rigor, clarity, and clinical relevance of our work. We hope our responses have adequately addressed your comments and that our revised manuscript is now suitable for publication in PLOS ONE.

Sincerely,

Se Joon Woo, MD, PhD

Department of Ophthalmology

Seoul National University Bundang Hospital

82, Gumi-ro 173beon-gil, Bundang-gu, Seongnam, Gyeonggi-do, South Korea, 13620

Tel: +82-31-787-7377

Email: sejoon1@snu.ac.kr

---

## [Decision Letter · Decision Letter 2]

11 Feb 2026

Intravitreal injections of corticosteroid and the risk of central serous chorioretinopathy

PONE-D-25-57894R2

Dear Dr. Se Joon Woo,

We’re pleased to inform you that your manuscript has been judged scientifically suitable for publication and will be formally accepted for publication once it meets all outstanding technical requirements.

Kind regards,

Alon Harris

Academic Editor

PLOS One

Additional Editor Comments (optional):

Reviewers' comments:

Reviewer's Responses to Questions

**Comments to the Author**

Reviewer #1: (No Response)

Reviewer #2: All comments have been addressed

2. Is the manuscript technically sound, and do the data support the conclusions?

Reviewer #1: Yes

Reviewer #2: Yes

3. Has the statistical analysis been performed appropriately and rigorously?

Reviewer #1: I Don't Know

Reviewer #2: Yes

4. Have the authors made all data underlying the findings in their manuscript fully available?

Reviewer #1: Yes

Reviewer #2: Yes

5. Is the manuscript presented in an intelligible fashion and written in standard English?

Reviewer #1: Yes

Reviewer #2: Yes

Reviewer #1: (No Response)

Reviewer #2: I have reviewed the latest revised version of the manuscript. The authors have addressed my previous comments satisfactorily, and the additional revisions made in response to the other reviewer’s comments have further improved the manuscript.

I continue to recommend the manuscript for acceptance for publication.

**Do you want your identity to be public for this peer review?** For information about this choice, including consent withdrawal, please see our For information about this choice, including consent withdrawal, please see our Privacy Policy .

Reviewer #1: No

Reviewer #2: No

---

## [Editor Report · Acceptance letter]

PONE-D-25-57894R2

PLOS One

Dear Dr. Woo,

I'm pleased to inform you that your manuscript has been deemed suitable for publication in PLOS One. Congratulations! Your manuscript is now being handed over to our production team.

Kind regards,

on behalf of

Dr. Alon Harris

Academic Editor

PLOS One